# Shifting Paradigm from Gene Expressions to Pathways Reveals Physiological Mechanisms in Blood Pressure Control in Causation

**DOI:** 10.3390/ijms24021262

**Published:** 2023-01-09

**Authors:** Alan Y. Deng, Annie Menard, David W. Deng

**Affiliations:** Research Centre, CRCHUM (Centre Hospitalier de l’Université de Montréal), Department of Medicine, Université de Montréal, Montreal, QC H2X 0A9, Canada

**Keywords:** blood pressure, congenic knock-in genetics, polygenic hypertension, epistasis, *FNDC1*

## Abstract

Genetics for blood pressure (BP) in human and animals has been partitioned into two separate specialties. However, this divide is mechanistically-misleading. BP physiology is mechanistically participated by products of quantitative trait loci (QTLs). The key to unlocking its mechanistic mystery lies in the past with mammalian ancestors before humans existed. By pivoting from effects to causes, physiological mechanisms determining BP by six QTLs have been implicated. Our work relies on congenic knock-in genetics in vivo using rat models, and has reproduced the physiological outcome based on a QTL being molecularly equal to one gene. A gene dose for a QTL is irrelevant to physiological BP controls in causation. Together, QTLs join one another as a group in modularized Mendelian fashion to achieve polygenicity. Mechanistically, QTLs in the same module appear to function in a common pathway. Each is involved in a different step in the pathway toward polygenic hypertension. This work has implicated previously-concealed components of these pathways. This emerging concept is a departure from the human-centric precept that the level of QTL expressions, not physiology, would ultimately determine BP. The modularity/pathway paradigm breaks a unique conceptual ground for unravelling the physiological mechanisms of polygenic and quantitative traits like BP.

## 1. Introduction

### 1.1. Background and Issues

Among the hidden conditions, hypertension is the most prevalent for the recent hospitalization of COVID-19 patients (https://www.cdc.gov/mmwr/volumes/69/wr/mm6915e3.htm, accessed on 17 April 2020). This disturbing risk has hastened our actions in discovering the mechanisms of hypertension pathogeneses. Human hypertension is set to have a systolic blood pressure of 140 mmHg and diastolic blood pressure of 90 mmHg (https://www.who.int/news-room/fact-sheets/detail/hypertension, accessed on 31 December 2021), where the severity is proportional to increases above these values.

Recently, genome-wide association studies (GWASs) have probably localized > 900 human quantitative trait loci (QTLs) for blood pressure (BP) by single nucleotide polymorphisms (SNPs) [1]. As it stands, irreplaceable and critical evidence is still missing that a given GWAS SNP can directly alter BP in vivo by causality. Because of this limitation, our understanding of a pathogenesis for human polygenic hypertension from GWAS alone is no more advanced than before its onset [2].

To surmount the functional deficiency of human GWAS, experimental knock-ins using inbred rodents have revealed the causative mechanisms from QTLs in physiologically regulating BP [3,4,5]. This is because a physiological change in BP is the prime mover and genesis driving the rodent knock-in genetics. In contrast, BP physiology and mechanisms are an afterthought for GWAS [6].

### 1.2. BP QTLs in Mammalian Ancestry

Across mammalian orders, rodent BP QTLs, as physiological proxies for humans, are self-evident, albeit often overlooked, because studying them in rodents is equivalent to revealing the same mechanisms in humans embedded in their common ancestors. Evolutionarily, humans and rodents, along with most land living mammals, possess a similar range of blood pressures [7], despite differences in other physiological characteristics (e.g., size). The only path for this to occur is that key mechanisms controlling BP must have been formulated, and stabilized ever since in the common ancestors of rodents and humans before they diverged > 90 million years ago (www.timetree.org, accessed on 31 December 2021). Thus, BP as a quantitative and polygenic trait has its genetic and mechanistic foundation deep in mammalian ancestry [3,4,5], long before humans came into being. Identifying QTLs for BP is central in understanding this foundation.

### 1.3. A QTL Is Molecularly Equal to One Gene

A QTL can be defined genetically by association/linkage, or by chromosome knock-ins [8]. Before molecularly known, a QTL broadly refers to a locus residing in a chromosome segment containing multiple genes. However, a QTL per se is one gene in a polygenic context, not a combination of genes, when molecularly identified to alter BP physiology in causation [8]. For instance, *C17QTL1* on rat chromosome (Chr) 17 is the one and single gene encoding *Chrm3* [muscarinic cholinergic receptor 3 (M3R)] [9,10,11,12]. *C17QTL1*/*Chrm3* alone has a major effect on blood pressure. Steeped in mechanism, one QTL is unambiguously equal to one gene that affects BP by causality and acts in a Mendelian mode [8,9]. In a semantic twist, some refers to a QTL molecularly identified as a QTG (quantitative trait gene).

Regardless of the terminology, understanding a QTL follows the same path of fathoming a gene. Before molecularly known, a gene was understandable as a Mendelian unit responsible for inheritance. Currently, we realize that a gene is a stretch of DNA encoding a protein product, which in turn, can impact the physiology of a trait in question (e.g., blood pressure). If one keeps referring to a gene purely as an antiquated Mendelian unit today, the concept would be inadequate. The same principle applies to a QTL. Namely, viewing a QTL as a segment of a chromosome is not only inaccurate, but insufficient for understanding it molecularly and mechanistically. The current proven principle is 1 QTL = 1 QTG = 1 gene [9,10,11,12].

Even in QTL mapping, the most appropriate description should be that a QTL is localized to reside in a segment of a chromosome. A QTL is not equal to a chromosome segment. One QTL = one QTG = one gene is what a QTL is in chemistry. Whether or not a QTL is molecularly identified does not change its molecular nature. Nevertheless, our recent understanding of 1 QTL = 1 QTG = 1 gene is fundamental in QTL research because it leads to mechanistic insights [9,10,11,12].

### 1.4. Mechanistic Unity in Regulating Mammalian Blood Pressure by QTLs

In vivo studies have begun to unify the BP QTLs of humans and rodents into a conceptual framework known as QTL modularity [3,4,5,13], which forms a mechanistic basis in physiological BP controls in causation [6]. In this context, a human QTL has a major physiological effect on BP [3,4,5]. The intergenic GWAS SNP close to *CHRM3* [14] is a fortuitous marker for a human QTL close by, not the QTL itself, because depleting it has no effect on BP [9,11]. Consequently, whether or not such a ‘common’ SNP is replicated by additional GWAS is an epidemiological undertaking [1], irrelevant to its physiological role on BP [15].

These counter-intuitive and unorthodox results contradict the predominant and human-centric doctrine known as the ‘omnigenic’ hypothesis [2]. It prescribes, with no physiological proof in causation, that a non-coding GWAS SNP in a polygenic context ought to quantitatively regulate QTL expressions to display ‘miniscule’ effects without physiology, and to cumulatively achieve BP.

A mechanistic shift in paradigm [16] is gradually unfolding from the popular ‘omnigenicity’ based on effects [2] to the little-recognized QTL modularity based on causality to BP changes [3,13]. While the soundness of human GWAS on BP has been widely accepted [1], a recognition of the modularity concept still awaits extensive validations with uncontestable in vivo evidence. We are an individual-researcher lab with limited resources and can only conduct experiments on testing QTL modularity in progressive phases [3,4,5,15].

### 1.5. Objectives

Currently, we expanded the rat QTL coverage to assess previously-unexplored human GWAS genes as a test of reproducibility of mechanistic outcomes from QTLs. Our new data have validated and broadened the paradigm of QTL modularity to comprise both rodents and humans as pathogenic pathways to polygenic hypertension [6]. Formerly-unsuspected components of these pathways have been implicated. Hopefully, our published work [3,4,5,15], and the current reinforcement of them, will encourage other scientists to extend our findings in animal models and humans. In this way, the validity of this evolving paradigm can be sweepingly tested. After all, the rarely-noticed Mendelism became universally appreciated and established after a 35-year indifference, only when Mendel’s experiments were broadly reproduced [17].

## 2. Results

### 2.1. Congenic Knock-In Genetics Is a Proxy in Functionally Mediating a Physiological Jump from Statistical Associations to Causation in Human BP Control

Conventional genetic coverage with statistics has been greatly expanded with GWAS into humans. However, statistics is not physiology and association is not causation. To make a qualitative leap from association/statistics into physiological mechanisms by causality, we have to resort to gene targeting on specific genes. Congenic knock-in genetics is a functional intermediate between them in identifying suitable gene candidates.

The principle of congenic approach is the same as the classical ‘knock-in’ of a single nucleotide [8]. It merely varies in a genome size and is applied as congenic knock-in genetics (Figure 1). When BP changes as a result, a QTL is physiologically defined in a chromosome region by causation [8]. In order to detect this change, our congenic knock-ins were carried out in the Dahl sat-sensitive rats (DSS) genetic background that had lost its genome buffering capacity in averting BP fluctuations [18] and in hypertension suppression [19,20,21].

The inbred DSS model is noted for its ‘salt-sensitivity’ for physiological studies. Despite this, the human GWAS genes that have been surrogated by DSS also function in general human populations, regardless of salt sensitivity [3,4,5,15]. Specifically, the DSS M3R signaling pathway is pro-hypertensive, even under a low salt diet [9]. High salt simply quickens hypertension and our studies of it by employing this model. Thus, DSS BP-regulating mechanisms are relevant to those of human essential hypertension in general populations, irrespective of the salt content. Among the DSS chromosome regions established to contain BP QTLs [13], only those matching human QTL signals from GWAS [1] were assessed here (Appendix A).

### 2.2. Establishing Each QTL/GWAS Gene in Directly Causing BP Changes in Physiology

Fractionating total variance gives an impression that one human GWAS SNP, and presumably one QTL marked by it, appears to show a ‘miniscule’ effect [1]. What is the importance in seeking six ‘drops’ of GWAS genes/QTLs (Table 1, Figure 1) in the ocean of >900 GWAS genes [1]? An informative answer came from identifying the true magnitude of physiological effect for each of these six QTLs. Each of them had a physiological BP effect of at least 43% alone (Table 1), in homogeneity, by causality, and under a uniform environment (Table 1 in reference [23] had more).

Here, a distinction is instructive between a statistical fractionation from the total variance and an actual physiological impact for each QTL. This is because the estimated BP effect for each of these 900 human QTLs in GWAS [1] was intermixed in heterogeneity from other genes in the whole genome, and compounded by uncontrollable environmental forces in the study populations. Due to this intractable uncertainty, an effect calculated from total variance does not prove whether the GWAS signal itself, and alone, can have a functional impact on BP in causation, let alone its magnitude of physiological effect. *C1QTL1* on DSS rat Chr 1 underscores this problem (Figure 1A).

*C1QTL1* was not mapped statistically, and explained 0% total variance in a heterogeneous rat population [24]. In contrast, by in vivo causation and under a uniform environment, *C1QTL1* alone physiologically altered BP by 52% (Table 1) and no combination with another QTL is needed to achieve this outcome. The calculation is as follows. The knock-in genetics defining *C1QTL1* by the C1S.L4 congenic strain decreased BP by 43 mmHg (Figure 1A). The total BP difference between DSS and Lewis rats was 83 mmHg (i.e., 178 mmHg for DSS minus 95 mmHg for Lewis). Thus, the physiological BP effect of *C1QTL1* was calculated as 43/83 = 52%.

Similarly, each of the remaining five QTLs singularly possessed a physiological BP effect ranging from 43 to 72% (Figure 1, Table 1), even though the fragment harboring each QTL had more than one human GWAS gene ortholog. Cumulatively, the six QTLs by themselves (Table 1) are physiologically redundant to account for the total BP difference between the two parental strains. If the ‘omnigenic’ hypothesis was physiologically valid, eliminating one QTL/GWAS gene among a multitude should have an imperceptible outcome on an organism. This was not the case, since *Chrm3* is solely responsible for *C17QTL1*, and abolishing it alone physiologically reduced BP by at least 50% in the BP difference between *Chrm3^+/+^* and *Chrm3^−/−^* [9].

Thus, the pivotal enigma to solve is not how ‘miniscule’ the BP effect from each GWAS gene would have, but instead, why there is such a surplus of GWAS genes that are physiologically excessive in modulating BP. Aggregating them cannot be justifiable, since that would impossibly lower the BP of rat/human to below zero, whereas fractionating them from the total variance is not physiological. Therefore, a well-founded physiological answer in vivo is required for their combined effects on BP.

### 2.3. Humans and Rodents Utilize a Common Mechanistic Plan of QTL Modularity in Controlling BP by Causality/Physiology

Intervals harboring *C18QTL3* and *C18QTL4* separately contain three and five human GWAS gene orthologs (Figure 1B). When they were combined by knock-in genetics into a ‘double’ congenic strain of C18S.L2, their combined BP did not surpass that of *C18QTL3* alone (Figure 1B). A 2 × 2 ANOVA [3,13] showed epistasis between the two QTLs (*p* < 0.002). Epistasis means one QTL masking the effect of another and occurs regardless of the number of GWAS genes involved. Thus, they belong to epistatic module 1 and the same pathway leading to the BP control [25]. *C18QTL2* is also in the same pathway [18] and the segment harboring *C18QTL2* bears 20 human GWAS gene orthologs (Figure 1B). The phenomenon of non-cumulativity [6] is evident, irrespective of the exact number of QTLs present.

The regions containing *C1QTL2* and *C1QTL3* carry 73 human GWAS gene orthologs together (Figure 1A). When merged (Figure 1A), the two QTLs showed the same BP as a singular QTL alone (ANOVA *p* < 0.001), thus classifying them in the same epistatic module. Since *C1QTL2* is in the same epistatic module 2 as *Chrm3* [9], single genes responsible for *C1QTL2* and *C1QTL3,* respectively, are likely to belong to the M3R signaling pathway and each participates in a different step within the pathway.

In total, 101 human GWAS genes reside in regions harboring the six QTLs by functional proxy (Figure 1). One QTL was assumed to correspond to one GWAS gene as *Chrm3* alone encodes *C17QTL1* [3,9]. At least six of these GWAS genes may function physiologically via two independent pathways in determining BP. BP is physiologically additive between two members of two separate pathways [13], and is the basic mechanism of QTL actions [6].

### 2.4. Quantity of QTL Alleles and Levels of Gene Expressions Are Not Important for QTLs’ Effects on BP by Causality/Physiology

Since the level of gene expression is a cornerstone upholding the ‘omnigenic’ hypothesis [2], we tested the dose response of normotensive alleles from three QTLs on rat Chr 1 that all decreased BP. Mean arterial pressures (MAP) (142 mmHg ± 4, *n* = 6) of heterozygous rats having one copy of the normotensive *C1QTL1* allele were not significantly different (*p* > 0.46) from the MAP (134 mmHg ± 3, *n* = 6) of homozygotes having two copies. Likewise for *C1QTL2*, MAPs [132 mmHg ±6 for heterozygotes (*n* = 5) vs. 139 mmHg ± 4 for homozygotes (*n* = 5)] were not significantly different (*p* > 0.54). Similarly for *C1QTL3*, MAPs [131 mmHg ±6 for heterozygotes (*n* = 5) vs. 125 mmHg ± 4 for homozygotes (*n* = 8)] were not significantly different (*p* > 0.67). Thus, the gene doses, and by inference, the levels of expression for the three QTLs, have no impact on diminishing BP in vivo by causality.

### 2.5. Associating the Consequential Effect of C1QTL1 in GWAS Fortuitously Landed a SNP Close to FNDC1 as a Functional Candidate for Causing Human Essential Hypertension

Identifying *CHRM3* as *C17QTL1* is the basis for genetically revealing a component as a step in a pathway [9,10], not only in DSS rats, but also for humans [3]. The functional dose of M3R signaling, not the gene dose of *Chrm3* expression, controls BP via M3R [11]. Supported by the evidence in the previous section, mechanisms modulating gene expressions by regulatory regions are not the primary focus in BP QTL discovery. Neither are non-coding SNPs, because the human GWAS non-coding SNP per se and rodent non-coding SNPs have no effect on BP [15], and merely flagged the *CHRM3* nearby [9]. Since *Chrm3* carries a function-altering mutation [9], missense mutations may directly change a pathway, and are priority, although not exclusive, choices for identifying candidate genes for following QTLs.

The interval containing *C1QTL1* carries nine genes in total [26]. *Fndc1* was the strongest function candidate among them for *C1QTL1* in DSS rats, because it carries multiple missense mutations (Appendix A). The congenic knock-in of *Fndc1* physiologically reduced BP on a high salt diet (Figure 1A). A human intergenic GWAS SNP, rs449789, was closest to *FNDC1*, 5 kb away [1]. Together, the causality of rat BP physiology by congenic knock-in genetics and the association of *FNDC1* from human GWAS coincide to support the potential importance of *FNDC1* as a QTL for the two orders of mammals.

We next evaluated the BP effect of *C1QTL1*/*FNDC1* on a low salt diet. The averaged MAP of the C1S.L4 congenic strain (Figure 1A) was 117 mmHg ± 2 (*n* = 5) and that of DSS parental rats was 130 mmHg ± 2 (*n* = 10). The difference between the two MAPs was significant (*p* = 0.002). Diastolic and systolic pressures were consistent with their MAPs. Thus, *C1QTL1*/*FNDC1* impacts on BP regardless of the salt content and is applicable to human essential hypertension.

rs449789 is naturally ‘knocked out’ in the rat (Table 2). The SNP cannot have an impact on the functionality of *C1QTL1*/*FNDC1* (Figure 1B), since BP was reduced without it, and the BPs of human and rats were comparable. Beyond the SNP, the genome sequence up to 1 kb around rs449789 was not conserved with that of the rat and mouse. The SNP started to appear as Simians diverged into Old World monkeys (e.g., in Rhesus macaques) and the ancestors of apes, and continued into Great Apes (Appendix A).

In contrast to an indifference of gene dose to BP and a lack of the non-coding SNP conservation, *FNDC1* coding regions are conserved by 89% between humans and rats. Physiologically, C1QTL1/FNDC1 should both be conserved between the rat and human as well as capable of potentially changing BP by function. To support this, five missense mutations in *Fndc1* codons altered its protein structure, and one of them was predicted to be damaging in function in DSS rats [26]. Furthermore, the consensus FNDC1 protein was conserved by 85% between rats and humans (Appendix A). Thus, the structure of FNDC1 is the strongest functional candidate for human *C1QTL1*. Finally, the functional directionality of *Fndc1* appears anti-hypertensive [26]. Probable 63 missense mutations were found in human *FNDC1* [27] (Appendix A).

### 2.6. The Broad C1QTL3-Residing Region Contains Multiple Functional Orthologs as QTL Candidates for Human GWAS Genes

A total of 58 GWAS genes in this region can be grouped on seven different human CHROMOSOMES (CHRs) (Appendix A). Thus, there are, at minimum, seven distinct QTLs physiologically affecting BP in the interval now tentatively designate as *C1QTL3* alone. The term QTL defined by congenic knock-in genetics is a functional gauge whereas a GWAS gene is not. Thus, a QTL was followed.

Although fine congenic resolution for each of the seven QTLs is not available at the present to the extent as defining *C1QTL1* (Figure 1A), congenic knock-in in C1S.L6 involves 17 human GWAS gene orthologs and decreased BP. Thus, at least one of these 17 orthologs is a new QTL, *C1QTL4*, even without an apparent functional candidate bearing missense mutations. The presence of this QTL is consistent with QTL1b named by other investigators [28].

A total of 19 GWAS genes were on human CHR11 from *GAB2* down to *SOX6* (Appendix A). *RELT* emerged as a functional candidate among them carrying missense mutations in the DSS ortholog (Appendix A). This new QTL, with *RELT* as the functional candidate, was designated as *C1QTL5RELT*, pending proof from congenic knock-in genetics. Likewise, two potential new QTLs are *C1QTL6UMOD* on CHR6, and *C1QTL7BAG3* on CHR10, respectively. Three GWAS genes within 560 kb of one another, *CBWD1, DOCK8,* and *KANK1* on CHR9, all carried missense DSS orthologs. The new QTL marked by them was named *C1QTL8CDK* (Appendix A). It is likely that more QTLs will emerge from them because 1 QTL = 1 gene and not all GWAS genes are QTLs.

Intergenic and intronic GWAS SNPs that spotted the adjacent QTLs by chance are absent in rodents (Table 2), and are products of primate evolution (Appendix A). These SNPs have no impact on BP, as discussed in the previous section. Probable missense mutations in each of these six rat orthologs as functional candidates for QTLs were detected (Appendix A).

#### QTLs on DSS Chromosome (Chr) 18 [25] Have Multiple Positional Orthologs from the Human GWAS [1]

The regions lodging three QTLs on rat Chr 18 (Figure 1B) contained 28 human GWAS genes, although no functional candidate was identified for lacking missense mutations (Appendix A). *NEDD4L* is not a functional candidate for *C18QTL3*, because no genetic variations were found that can change its function or expressions [29]. No coding mutations have been detected among the widely-used rat strains. Since a great degree of homology exists among the seven members of *PCDHA* genes, it is not clear whether any of these missense mutations was specific for that gene of the gene family, or if the comparison was made between differing members of the same gene family.

## 3. Discussion

### 3.1. Principal Results from This Study

(**a**) By emphasizing causes instead of their effects, we have shown shared physiological mechanisms of BP control from common ancestors of humans and rodents. (**b**) Specifically, six distinct QTLs from inbred DSS rats revealed mechanistic causes in BP regulation for at least six human GWAS genes. They together have redundant effects on BP, because they likely function in two separate pathways. Previously-hidden components of these pathways have been implicated from candidate genes representing these QTLs. (**c**) The non-coding SNPs marking these six QTLs/human GWAS genes are byproducts of primate evolution unrelated to BP regulation. These SNPs are human-centered and inscribe adjacent QTLs, rather than being QTLs per se.

### 3.2. Importance in Studying QTLs by Causality/Physiology

Distinguishing physiological causes from their consequences yields mechanisms underlying a polygenic trait. GWAS firmly stands on its own merit as a branch of population genetics [2,30]. Its values in human epidemiology have recently been favored in analyzing phenotypic variations in heterogeneous populations. GWAS is not burdened by causes of physiological mechanisms and focuses instead on analyzing their consequences. This is similar to Newton’s eloquent descriptions of the effect of gravity as a domain of physics so long as the cause of gravity is off limit (www.britannica.com/science/general-relativity, accessed on 31 December 2021).

Why does one bother with the nuanced separation between cause and effect for the same phenotype in quantitative and polygenic genetics? This is because the fundamentals of causes and effects produce completely different predictions and explain contradictory phenomena. When causes are front and center, the human and animal results presented in this paper make mechanistic and physiological sense in regulating BP [6], but from the vantage of epidemiological effects [2], they do not. Similarly, if one views gravity from the Newtonian descriptions of effects, the ‘erratic’ orbit of Mercury, origin of the universe and black holes are nonsensical. If you view the same phenomena from the causal view of curvature in warped space time, they all fall into place (www.britannica.com/science/general-relativity, accessed on 31 December 2021). Adopting the same logic, the following discussions highlight some salient features directly addressing BP as a polygenic trait in causation/physiology.

### 3.3. Association/Statistics Is Not Causation/Physiology

Recently, an ‘omnigenic’ hypothesis has been expounded to explain GWAS results from an effect’s angle [2]. This can be attributed to an anthropocentric (or human-centered) hypothesis because non-coding GWAS SNPs only exist in humans, but not in rodents, as our previous [3,4,5,15] and current data have shown. It essentially proposes that regulations of gene expressions at the cellular level would explain the roles of GWAS SNPs in human polygenic traits including BP [2]. However, physiological studies in vivo are not replaceable by in vitro cellular studies because of an uncoupling between the systemic BP and cell/tissue activities (discussion 4.2.1. in reference [15]).

Our current phenotyping directly compared a change in BP by a QTL (Table 1). In contrast, GWAS uses the total variance to estimate the effect of a SNP on BP. Total variance is the expected value of the squared deviation of a random variable in BP from the population mean [30]. The total variance essentially measures the extent of BPs spreading out from their average value in outbred populations [30]. The basic GWAS design [2] is to associate genome markers with unknown functions to total variance in BP. Two limitations are apparent from GWAS in finding the genetic causes of BP control.

First, genome markers chosen for GWAS [2] are entirely based on their prevalence in study populations, which have nothing to do with BP control. If a GWAS SNP actually marks a BP QTL nearby [2], this coincidence arrives purely by chance, and is not driven by a change in BP.

Second, a genetic variation in controlling BP will cause variability in total variance. The opposite is not true (i.e., not all total variance is caused by alterations in genetic elements regulating BP). In fact, the leading contributions to total variance are environmental variations, not alterations in the mechanistic physiologies of QTLs [8]. For instance, the blood pressure total variance in inbred DSS rats was considerably enlarged solely due to a change in dietary salt [9,31], while the genetic variability was nearly zero and other environmental factors were uniform.

Thus, GWAS statistically identified a SNP associated with BP [2] as a genome entry point into identifying a cause of variability. Whether or not it marks a real QTL can only be resolved by a physiological proof in causation. Examples are our current and previous studies [3,4,5,9,10,11,12,15]. A case of invalidation can be shown by the gene encoding inducible nitric oxide synthase (*Nos2*). A non-coding marker in *Nos2* was statistically associated with BP in a rodent heterogeneous population [32]. However, upon physiological testing in causation, lacking a blood pressure effect proved that the statistical significance in *Nos2* was actually false positive in causation/physiology [33].

### 3.4. Independence of Gene Dose from a QTL on BP in Causation/Physiology

BP of *Chrm3^+/−^* with one functional copy of *Chrm3* was the same as that of *Chrm3^+/+^* with two copies [9,11]. This is because one copy of the M3R protein sufficiently enables its signaling pathway to function normally as a cause to BP control. One dose of a normotensive QTL allele lowers BP to the same extent as two doses for *C1QTL1*, *C1QTL2*, and *C1QTL3* (Section 2.4). This heterozygous dominance for the three QTLs plus *C17QTL1*/*CHRM3* indicates that each of them behaves as if it were a typical Mendelian gene. Other DSS QTLs [34] function in similar gene-dose independence in the physiology control of BP. This line of evidence does not conform to the central idea of the ‘omnigenic’ hypothesis that the gene expressions may determine blood pressure without physiology.

### 3.5. The Effect of a GWAS Gene on Blood Pressure Is Not Miniscule in Causation/Physiology

Assuming that a GWAS SNP can mark a BP QTL nearby, its effect on BP from the GWAS is not based on how much it can directly change BP physiologically, as conducted in the current work (Table 1). Instead, the effect of this GWAS SNP was fractionated for total variance [2] (i.e., the extent of blood pressure spreading out from their average mean in outbred populations) [30]. In so doing, this SNP seemed to exhibit a miniscule amount [2], deviating from the average value of blood pressure and not a physiological alteration in the blood pressure itself.

The true blood pressure effect of a QTL in vivo has to be assessed by measuring the physiological change in BP per se in isolation, while keeping the rest of the genome homogeneous and the environment uniform. It turns out that the magnitude of the physiological effect from a QTL is at least 43%, which is not miniscule (Table 1). When a QTL is molecularly identified, the BP of *Chrm3*-nulls was >50% lower than the *Chrm3* wild type in causation/physiology [9]. Since the pathway is the physiological cause in BP control, the *Chrm3*-nulls inactivated the entire M3R signaling pathway and the BP dropped as a result. Animal model manipulations testing human GWAS results are elaborated further in the discussion in [3].

### 3.6. Principal BP-Regulating Mechanisms in Humans Originated in Common Mammalian Ancestors and Are Not Due to Convergent Evolution

Modularity as a basis for physiological BP control [9] is consistent with evolutionarily-conserved BP regulating pathways rooted in the common ancestors of humans and rodents. This is because humans, rodents, and most land-living mammals have similar blood pressures [7]. The human-centric non-coding GWAS SNPs that gave rise to ‘omnigenicity’ [2] would nullify these conserved pathways, since these SNPs only started to emerge in primates, but are not present in rodents (Table 2 and Appendix A).

Conversely, the rodents’ non-coding *Chrm3* SNPs are not present in humans. Eliminating the M3R signaling pathway did not involve any of them, but BP changed [9]. Thus, functional M3R signaling coexists in humans and rodents in determining the hypertension pathogenesis, to which non-coding rodent *Chrm3* SNPs and the human-centered GWAS SNPs are irrelevant [15]. These SNPs coincidentally label probable physiological QTLs nearby [3,4,5,15]. These SNPs per se might have various functions in controlling gene expressions, epigenetics, and/or even be eQTLs in cell-based studies [1]. They contribute to nearly 0% of the overall mammalian physiology of BP controls including humans [15]. The appearance of human non-coding SNPs is a genome-wide occurrence in primate evolution that has nothing to do with BP regulations.

The M3R signaling pathway exemplifies the existence of a BP-controlling mechanism in common mammalian ancestors [15] long before humans made an appearance, and explains similar BPs among humans, rodents, and other mammals. Obviously, environments differed during the evolution of these mammals in the past 90 million years (www.timetree.org, accessed on 31 December 2021), and modern humans and wild rodents live in different surroundings. Convergent evolution through different mechanisms implicated by human-centric SNPs [1] cannot produce similar blood pressures by accident to those of other orders of mammals. The only way to achieve similar BPs among them is that the key mechanisms regulating BP must have been established in common ancestors of mammals, before they evolutionarily diverged.

### 3.7. A Wide-Ranging Scope of Modularity in BP Control Based on Causation/Physiology

Certain QTLs [6,9] begin to function at embryogenesis, before adult BP physiology even starts. This is consistent with the idea of modularity involved in BP control, because a pathway can temporally begin at embryogenesis and continue on to adulthood [9]. The ‘omnigenic’ view entirely relies on adult BP variations to the exclusion of embryogenesis.

The modularity paradigm explains additional cause-based phenomena that the effect-based ‘omnigenicity’ would consider out of place. They include a regulatory hierarchy in a pathway between two BP QTLs [23], a higher echelon in BP regulation leading to hypertension suppression [19,20,21], monogenic hypertension at the same step of a pathway versus polygenic hypertension in different steps of a pathway [6], the hypertensive state not caused by the same mutated QTL in a polygenic context [3], a paradox of a ‘common’ SNP with no effect on BP identifying a ‘rare’ BP-impacting variant nearby [3], and pathogenesis-based personalized medicine [6].

QTL modularity is the genetic framework in physiologically modulating BP embedded in ancestral genomes. Six human GWAS genes/QTLs from current studies implicate two pathways of hypertension pathogeneses including the M3R signaling pathway [9,10]. This pathway conservation is the physiological bedrock for having similar BPs between humans and rodents [7]. Because of this, the basic framework of BP-controlling mechanisms in pathways with multiple steps must have been established in common ancestors of mammals before they started to diverge [3,4,5,15]. Nevertheless, this conservation does not exclude a later appearance of new BP-controlling mechanisms during mammalian evolution. However, any genome development unique to humans cannot offset the ancestral BP-physiology that have been formulated from the underlying mechanisms based on QTLs.

### 3.8. Inbreeding Reveals Modularized Mendelism, Not Mixed Inheritance, as the Mechanistic Basis of Polygenicity of BP Regulation

The results presented in the current manuscript are based on physiological studies in vivo from inbred rodent strains, and seem uninterpretable by the quantitative genetics principle predicated on GWAS from outbred populations [2]. If inbreeds have a genetic architecture different from oubreeds, would polygenic inbreeds and outbreeds follow different physiological rules? In that case, the reach of ‘omniginicity’ [2] has become limited to only polygenic outbreeds, not polygenic inbreeds.

Alternatively, the assumed functions of human GWAS SNPs from outbreeds appear too intuitively automatic to require any in vivo proof in causation [2]. Estimates from sophisticated statistics are close enough and backed by functional assays in vitro, and seem to satisfy the hunger for physiological implications from GWAS [2], even though not the physiology mechanism itself. Outbred genetic architectures would be literally taken as an indirect display of physiology and the mechanisms of blood pressure regulation. After all, direct human studies are believed to be more ‘pertinent’ to drive medical applications in clinics than studying rodents. The GWAS’s importance seems to be further enhanced by shrouding their results in a mechanistic mystery as being ‘unknowable’ [2]. ‘Nitpicking’ inquiries into their mechanisms and causality may be ‘superfluous’. Or are they? Can we gain insights into BP-regulating mechanisms for outbreeds from analyzing inbreeds?

Obviously, inbreeding from heterogeneous populations have selected and then fixed a slice of genetic architecture controlling the BP physiology into an inherited, homogeneous, and polygenic trait. Inbreeding leads to establishing distinct and contrasting hypertensive and normotensive strains, which is equivalent to using a prism to separate light into a visual rainbow spectrum with differing wavelengths and frequencies. If one insists on solely studying outbreed populations epidemiologically, we could lump all of the genetic architectures together and would conclude a mixed inheritance with continuous variations [2]. Without inbreeding/prisming, we would not know how or if genetic architectures controlling BP in outbreed populations are actually composed of distinguishable and individual components. Each can be identified in causation/physiology to follow the Mendelian mode. It is only by studying inbreeds that the modularity of multiple QTLs in BP control has emerged [6]. In retrospect, if Mendel would have analyzed continuous and ambiguous phenotypes of peas from outbreed populations, he would never have formed his fundamental laws of heredity and debunked the theory of mixed inheritance.

### 3.9. Inferred Pathogenic Pathways for QTLs in Causation/Physiology

Pathogenic pathways of hypertension can be inferred from the molecular bases of QTLs. Since the QTLs in question (Figure 1, Table 1, Appendix A) have not been molecularly identified like *C17QTL1*/*CHRM3* [9], their roles in BP physiology are mostly inferred from the functional candidate genes with missense mutations representing them.

FNDC1 may be in a cascade of signaling mechanisms leading to BP control by modifying connexin 43 [26]. RELT protein is a member of the tumor necrosis factor (TNF)-receptor superfamily highly expressed in spleen, and is involved in immune T-cell activation in null mice [35]. UMOD is specifically produced in kidneys and is involved in a variety of diseases known as uromodulin-associated kidney disease. Both missense mutations and nulls show kidney phenotypes. Uromodulin has been shown to be pro-hypertensive in mice [36,37], without the involvement of the human GWAS SNP rs13333226, which mice naturally lack. BAG3 is a co-chaperone protein involved in removing bad proteins, and has a role in anti-cardiac myopathy [38]. *CBWD1* is ubiquitously expressed, but little studied (www.ncbi.nlm.nih.gov/gene/55871, accessed on 31 December 2021). DOCK8 is involved in nucleotide exchange in immunity. A loss of its function causes immunodeficiency [39]. KANK1 is a member of scaffold proteins bearing ankyrin repeats and plays a role in actin polymerization and cell motility [40]. Little is known about its role in BP physiology.

Probable mechanistic steps are unchartered with regard to connecting any of these proteins to the existing BP pathways 1 and 2 [13]. In pathway 2, the M3R signaling [9,11] is not the end-stage physiological step impacting on BP and the pathway itself suggests numerous steps, either upstream or downstream. The adrenergic pathway may lie down stream of it. It remains unexplored how these two pathways may be linked and what additional members are upstream of M3R, downstream of the adrenergic pathway, and mediating components in between.

### 3.10. Caveats and Limitations

First, possessing missense mutations in human GWAS orthologs per se does not genetically prove such genes to be the QTLs in question. The functional candidacy based on it merely provides entry points toward probable steps in two pathways made by each QTL. Viable gene-targeting of their codons can test their functionality on BP in DSS rats [9].

Second, structural mutations are not exclusive molecular bases that can affect a pathway in question. Seventeen GWAS gene orthologs in the *C1QTL4*-residing region (Section 2.6) had no missense mutations in DSS rats. Similar situations applied to the three regions on Chr 18 (Figure 1B). In spite of this, a BP QTL can be one gene encoding a member of a multi-component complex and can affect the function of the entire complex by altering the stoichiometry of its composition. As a result, the pathway as a whole may be impacted.

## 4. Materials and Methods

### 4.1. Animals

The study was conducted according to the guidelines of the Declaration of Helsinki, and approved by the Institutional Review Board (Comité institutionnel de protection des animaux, CIPA). Hypertensive inbred strain, Dahl salt-sensitive rats (DSS), was the basis for our genetic work.

### 4.2. Experimental Protocols

Breeding procedure, dietary treatments, telemetry implantation, postoperative care, and BP measurement durations were essentially the same as reported previously [3,4,5,15]. In brief, rats were males, which were weaned at 21 days after birth, fed a low salt diet for 14 days, then a high salt diet until the end of the experiment. Telemetry probes were implanted at 56 days after birth (namely 3 weeks after the high salt diet). In BP presentations (Figure 1), the averaged readings of mean arterial pressures (MAP) during the measurement were given for each strain.

### 4.3. Statistical Analysis

Repeated measures’ analysis of variance (ANOVA) followed by Dunnett’s test, which corrects for multiple comparisons and unequal sample sizes, was used to compare a parameter in MAP between two groups, as reported previously [13]. The power and sample size calculations in the analysis were the same as those given previously [3].

## 5. Conclusions

A new age is dawning on a classical revival in distinguishing causes from effects, and on restoring Mendelism to its rightful place as a pivotal hinge governing polygenic traits. Genetically, the polygenic trait, blood pressure, is created by modularized Mendelian ‘monogenic’ QTLs. In a sense, Mendelism to genetics is akin to the carbon element to chemistry. Graphite and diamonds are its ‘poly’-forms that are nothing but highly-‘modularized’ carbons covalently-bonded in different ways. Mechanistically, BP physiology is made by a pathway with ‘poly’-steps, and each QTL in the same module functions at a step within the pathway encoded by a Mendelian ‘monogenic’ gene. Multiple pathways are ingrained in mammalian common ancestors in determining BP. Unraveling the causative physiological mechanisms controlling BP proceeds alongside epidemiological surveys of their after-effects. In certain instances, such a cause and effect converge to the same QTL location, although proof of them are separate. This principle of modularity is an ‘out-of-bottle genie’ and lays a physiology-based foundation for polygenic and quantitative traits.

## Figures and Tables

**Figure 1 ijms-24-01262-f001:**
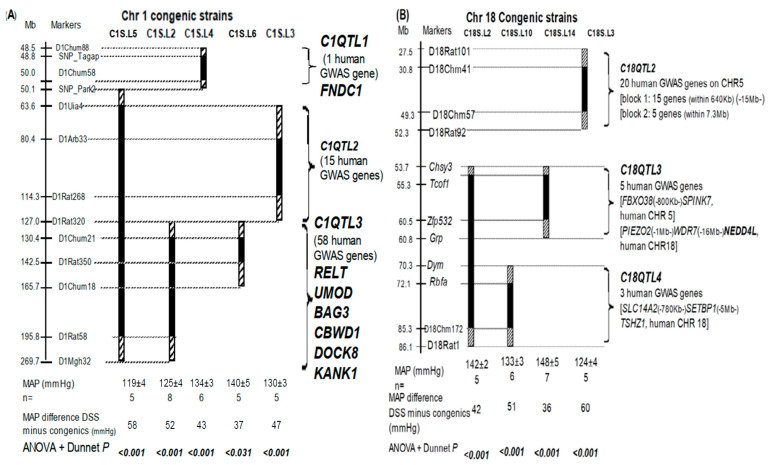
Congenic knock-in genetics defining chromosome intervals harboring BP QTLs in vivo in causation. Solid bars under congenic strains symbolize the hypertensive Dahl salt-sensitive (DSS) chromosome fragments that have been replaced by those of normotensive Lewis. Striped bars on the ends of solid bars indicate ambiguities of crossover breakpoints between markers. The full gene nomenclature corresponding to their abbreviations is given in the Table 1 legend. Mean arterial pressures (MAPs) for DSS and congenic strains are averaged for the period of measurement and are given at the bottom of the map. Significant *p* values are emphasized in bold and italics. ± indicates SEM. (**A**) Chr 1; (**B**) Chr 18.

**Table 1 ijms-24-01262-t001:** DSS QTLs with missense mutations functionally capturing human GWAS orthologs.

Rat QTL Name	Functional Magnitude BP Effect (Figure 1)	Rat Functional Candidate Gene	Human GWAS SNP (See Table 2)	Rat GWAS SNP Ortholog (See Table 2)	Closest Human Functional Gene	# Probable Human Coding Mutations
** *C1QTL1* **	52%	missensed *Fndc1* ^§^	rs449789	Non-existent	** *FNDC1* **	63
** *C1QTL2* **	57%	No candidate				
** *C1QTL3* **	63%	missensed ^§^ ***Relt*** ***Umod*** ***Bag3*** ***Cbwd1*** ***Dock8*** ***Kank1***	Multiple intergenic/ intronic SNPs	All non-existent except for 1 SNP at 3′UTR	** *RELT* ** ** *UMOD* ** ** *BAG3* ** ** *CBWD1* ** ** *DOCK8* ** ** *KANK1* **	6 6 8 8 20 14
** *C18QTL2* **	72%	No candidate				
** *C18QTL3* **	43%	No candidate				
** *C18QTL4* **	61%	No candidate				

QTLs and their BP effects are given in Figure 1. § shows confirmed mutations between our database and that of the public rat genome [22]; detailed data on coding mutations are presented in Appendix A. BAG cochaperone 3 (***BAG3***); COBW domain containing (***CBWD1***); dedicator of cytokinesis 8 (***DOCK8***); fibronectin type III domain containing 1 (***FNDC1***); KN motif and ankyrin repeat domains 1 (***KANK1***); RELT TNF receptor (***RELT***); uromodulin (***UMOD***). No candidate indicates no genes with missense mutations.

**Table 2 ijms-24-01262-t002:** A survey of sequence homologies between humans and rats for GWAS SNPs.

Human SNP/ Marked Gene	Rat Homology	Note
rs449789/*FNDC1* (intergenic)	No	Haphazard hits in 1Kb sequence used for blast; several mini- regions of 20–24 bp of homology randomly distributed on rat Chr1, but not in the right region.
rs7115605*/RELT* (intergenic)	No	Similar to above
rs13333226/*UMOD* (intron)	No	Similar to above
rs72842207/*BAG3* (intron)	No	Similar to above
rs2992854*/CBWD1* (intergenic)	No	Similar to above
rs643058*/DOCK8* (intergenic)	No	Similar to above
rs520015*/DOCK8* (intron)	No	Similar to above
rs604470*/DOCK8* (intron)	No	Similar to above
rs16923342*/KANK1* (intergenic)	No	Similar to above
rs60191654/KANK1 (intergenic)	No	Similar to above
rs17369029*/KANK1* (3′UTR)	Yes	197 bases hit (90.2%homology) in 1 Kb sequence used for blast on Chr1; SNP present, but the surrounding sequence in less conserved.

Gene names are given in the legends in Table 1. Appropriate sequence surrounding a SNP in question was blasted into the rat genome at: https://genome.ucsc.edu/cgi-bin/hgGateway, accessed on 31 December 2021.

## Data Availability

Data are available upon request to the corresponding author after publication.

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
