# Peer review of "Shifting Paradigm from Gene Expressions to Pathways Reveals Physiological Mechanisms in Blood Pressure Control in Causation"

_ijms, 2023, doi:10.3390/ijms24021262_

Round 1

Reviewer 1 Report (New Reviewer)

the manuscript by  Deng et al has tried to identify the the loci that are associated with hypertension in human by studying the candidate SNPs in male rats. the study is rather fine but I have the following comments 

- the title is not informative and it is very general. it looks like a review article rather than a research article. Please rephrase to a specific one that reflects the findings of this study. 

- the materials and methods should be stated briefly in the abstract. e.g. animals, study design etc....

- the introduction is too long and difficult to follow. language should be carefully revised. 

- It is unusual to make a separate paragraph for the objective in the introduction. 

-the QTL should be clearly defined in the first part of the introduction. 

- I acknowledge that the authors refer to their previous published research, but still they should mention the methodology appropriately in materials and methods section. 

- the paper by He et al, 2022, Rs420137, rs386360 and rs7763726 polymorphisms in fibronectin type III domain containing 1 are associated with susceptibility to coronary heart disease: Analysis in the Han population, Rs420137, rs386360 and rs7763726 polymorphisms in fibronectin type III domain containing 1 are associated with susceptibility to coronary heart disease: Analysis in the Han population, Front Cardiovasc Med., doi: 10.3389/fcvm.2022.964978. 

The above paper should be discussed as they report similar results.

-in page 8 lines 817-818 the authors citing a website ((www.britannica.com/science/gen- 318 eral-relativity). that is not opened. Please cite the information in a proper way.

-it is not usual to put limitations of the study in a separate paragraph. the use of the word '' Caveats'' in unusual as well. 

Author Response

please see attached response in a separate file

Reviewer 2 Report (New Reviewer)

The role of genetics in cardiovascular diseases approach is rapidly increasing. It highlights different orientations that will be useful in the management of diseases, particularly Hypertension sticking intervention on causality is needed which will significantly impact the prognosis for patients.

Author Response

We agree with your assessement.

This manuscript is a resubmission of an earlier submission. The following is a list of the peer review reports and author responses from that submission.

Round 1

Reviewer 1 Report

The topic of the manuscript concerns genetic pathways in the pathophysiology of hypertension and follows the previous works of the authors. All parts of the manuscript are adequate to the solved problem, it is appropriate to organize the manuscript in the form of Introduction, Objective, Methodology, Statistical analysis, Results, Discussion, Conclusion.

I have several comments:

In the Introduction part:

I recommend including the definition of hypertension and severity of hypertension.

Methods:

Despite the fact that the authors provide citations of their works in the methodological section, I recommend a brief presentation of methodological procedures in this work as well.

I recommend presenting the statistical analysis separately.

Discussion:

p. 10, lines 334 – 341: The authors refer to their previous works, and therefore I recommend summarizing the results of the given works in the text or in the form of a Table.

p. 12, section 3.9 „Inferred pathogenic pathways...“:

I recommend including explanation of the pathogenic pathways also in the graphic form (as Figure).

Reviewer 2 Report

The manuscript by Deng et al. “Shifting paradigm from gene expressions to pathways reveals physiological mechanisms in blood pressure control in causation“ is very difficult to read and understand. I will demonstrate this problem by commenting the abstract, sentence by sentence.

The abstract

“Genetic experimentations for blood pressure (BP) in human and animals have been partitioned into 2 separate specialties. However, this divide is mechanistically misleading.“

Comments

It is not possible to perform genetic experimentation in humans, genetic analysis of blood pressure regulation in humans just uses different methods compared to animal models. Specifically, GWAS in humans provide sufficient statistical power to identify SNPs associated with blood pressure variability while studies in rodent models selected for susceptibility to high blood pressure use linkage analyses in genetically segregating populations to identify QTL which are confirmed in follow-up experiments in congenic strains. Ideally, genes responsible for QTL are identified at the molecular level as variants of specific genes, so called QTG (quantitative trait genes). Both human and animal studies are usuful to understand the pathophysiology of hypertension. So it does not make sense that “…this divide is mechanistically misleading.“

“BP physiology is mechanistically initiated by quantitative trait loci (QTLs).“

Comments

This sentence does not make sense. BP physiology cannot be mechanistically initiated by QTLs. The authors maybe wanted to say that identification of genetic determinants of blood pressure starts by mapping of QTLs to specific segments of chromosomes, followed by their genetic isolation in congenic strains and sublines.

“The key to unlock its mechanistic mystery lies in the past with mammalian ancestors before humans existed.“

Comments

This sentence does not make sense. Maybe the authors wanted to say that basic physiological mechanisms regulating blood pressure are evolutionary conserved between animal models and humans.

“We hypothesize that humans and rodent share similar mechanisms.“

Comments

Such hypothesis is useless. Many physiological and pharmacological studies have shown that basic physiological mechanisms regulating blood pressure are similar in animal models and humans.

“By shifting the focus from epidemiological after-effects to physiological causes, we have identified physiological mechanisms determining BP by QTLs.“

Comments

I am not sure I understand the meaning of “epidemiological after-effects“; are they related somehow to GWAS? It is not possible to identify physiological mechanisms of blood pressure regulation by QTL mapping. QTL is a segment chromosome usually with multiple genes. Identification of a QTL provides information about location of putative genetic determinants (not specific genes) regulating blood pressure but provides no information about physiological mechanisms regulating blood pressure. Maybe authors wanted to say that physiological analysis of congenic strains with genetically isolated QTL might provide some hints about physiological mechanims related to blood pressure diffrence.

“The evidence has reproduced the outcome that each QTL genetically acts as a building block by Mendelian monogenicity.“

Comments

This phrase that “…QTL genetically acts…by Mendelian monogenicity“ is an oxymoron.

“A gene dose for a QTL is irrelevant to BP controls.“

Comments

A QTL is not a gene, thus “a gene dose for a QTL…“ does not make sense

“Together, QTLs join one another as a group in modularized Mendelian fashion to achieve polygenicity.“

Comments

According to the authors a group of QTLs is similar to a Mendelian trait and thus achieves polygenicity. It does not make sense.

“Mechanistically, the QTLs in the same module appear to function in a common pathway. Each is involved in a different step in the pathway towards polygenic hypertension.“

Comments

QTLs are segments of chromosomes with many genes. Thus it is not possible to analyze mechanisms of chromosome segments.

“This emerging concept is a departure from the human-centric precept that the level of QTL expressions, not physiology, would ultimately determine BP.“

Comments

According to the authors, GWAS are human-centric because they use common SNPs that are specific to humans and are not present in rodent models. The authors argue that these polymorphisms cannot represent genetic variants regulating blood pressure. However, this is misunderstanding, nobody  claims that common variants used in GWAS are reponsible genes. And of course, GWAS are human-centric.

QTLs are segments of chromosomes and therefore “QTL expressions“ is a nonsense.